# Slot-and-Frame Schemas in the Language of a Polish- and English-Speaking Child: The Impact of Usage Patterns on the Switch Placement

**Dorota Gaskins [1],\*, Ad Backus [2] and Antje Endesfelder Quick [3]**

[1] Department of Applied Linguistics and Communication, Birkbeck College, University of London, Malet Street, London WC1E 7HX, UK

[2] Department of Culture Studies, School of Humanities and Digital Sciences, Tilburg University, P.O. Box 90152, 5000 LE Tilburg, The Netherlands; A.M.Backus@uvt.nl

[3] Institute of British Studies, Leipzig University, GWZ, Beethovenstraße 15, 04107 Leipzig, Germany; antje.quick@uni-leipzig.de

\* Correspondence: dorota.gaskins@yahoo.co.uk

**Abstract:** How does the bilingual child assemble her first multiword constructions? Can switch placement in bilingual combinations be explained by language usage? This study traces the emergence of frozen and semi-productive patterns throughout the diary collection period (0;10.10–2;2.00) to document the acquisition of constructions. Subsequently the focus falls on most frequently produced monolingual and bilingual combinations captured through 30 video recordings (1;10.16–2;5.11) which are linked to the diary data to confirm their productivity. First, we verify that like in monolingual development, frequency-based piecemeal acquisition of constructions can be reproduced in our bilingual diary data: in the child's earliest combinations 87% are deemed as semi-productive slot-and-frame patterns. Second, video recordings show that productivity, understood as a function of type frequency, plays a role in determining the switch placement in early bilingual combinations only to some extent. A more accurate explanation for why frames from one language take slot fillers from another is their autonomous use and semantic independence. We also highlight limitations of input: while the child was raised with two languages separated in the input, she continued to switch languages which suggests that switching is developmental.

**Keywords:** bilingual; slot-and-frame; child codeswitching

## 1. Introduction

Study of children's early productions has been a focal point of usage-based research due to their potential to account for the journey a child's mind makes towards adult-like linguistic proficiency. Similarly, children's early language use has enjoyed considerable interest in research on bilingualism, as the simultaneous acquisition of two languages allows us to go into questions about how children learn to separate languages in their minds. Usage-based studies of bilingual acquisition, however, are rare, and our paper aims to contribute to a growing recognition that this gap needs filling. We will argue that both contributing research traditions stand to gain from considering bilingual acquisition data through a usage-based lens. Specifically, children's use of codeswitching (CS) in a setting in which parental input only contains negligible instances of the phenomenon will tell us something about the limitations of input characteristics in accounting for linguistic competence, while in no way denying that input is of crucial relevance. In addition, we will argue that studying children's CS from a usage-based perspective improves our understanding of dominance, an obvious but theoretically problematic concept in bilingualism studies.

### 1.1. Usage-Based Perspective on Bilingual Acquisition

Our view of language acquisition reflects constructivist models of linguistic representation which see linguistic competence as an inventory of form-meaning pairings (constructions) whose nature forever oscillates on the continuum from frozen unprocessed chunks, to partially schematic and eventually more abstract (Croft 2001; Goldberg 1995; Langacker 1987). Frozen constructions are multi-morphemic and multiword chunks which are acquired from the speech stream as wholes (e.g., *gimme* and *all gone*) and used as if they were single lexical items; children seemingly having no awareness of the parts (Peters 1983; MacWhinney 1978, 2014). Frozen items feature heavily in early speech: in children's first 50-word vocabularies, calculated as both single and multiword productions, on average 17.8% (range: 2–42%) and in their first 100-word vocabularies on average 21.2% (range 5–44%) of items are frozen multiword combinations (Lieven et al. 1992). The acquisition of frozen chunks provides a window into the development of schemas: when two words such as *All gone* are first used together, the child only becomes aware of a pattern with repeated opportunities to hear it in different configurations which allow for it to be segmented. As a result, one of these two words (e.g., *all*) is replaced with another (e.g., *mummy*, *daddy*), thus creating a slot *X gone* into which similar words from the child's linguistic repertoire can be inserted spontaneously at the moment of speaking. Such assembly necessitates categorical perception which allows for words to be selected and combined. Meanwhile, the word *gone* remains fixed acting as a pivot for this two-word combination.

Obviously, adult competence consists of more than just frozen forms, and indeed child data show early forms of productivity. This is visible in the second type of combination, called partially schematic constructions, which combine frozen and productive elements. In such multiword constructions, there are at least two frozen elements: one or more morphemes or words, and a pattern in which that word or morpheme (sometimes referred to as 'pivot') is a fixed element; the pattern is often referred to as a 'frame.' The open element in the pattern, often referred to as the 'slot', is subject to filling by whichever relevant word or morpheme helps conveying the intended meaning. The mechanism by which a speaker, be it a child acquiring the language or a fully functioning adult speaker, arrives at the activation of optimal slot fillers is not very well understood yet. However, we do know that they dominate early lexicons: using a combination of diary and video recorded data, Lieven et al. (1997) report that among the first 400 multiword constructions used by 11 monolingual toddlers (1;0–3;0) on average 60% (ranging from 51–72%) are such partially schematic patterns including *Put in X*, *I want to X* and *Go to X*. The time it takes to develop 25 patterns from the vocabularies of first 100 words ranges from 3 to 9 months (Lieven et al. 1997). Data from monolingual two-year-olds video recorded on a dense sampling schedule show that 78–92% of utterances can be classified as instantiating frames with open slots, with most slots filled with nouns and noun phrases and increasing in complexity as a function of increasing mean length of utterance (MLU) (Lieven et al. 2009). Later in acquisition and with mounting experience of language use, all parts of such utterances become fully processed and open to a broader range of elements they attract as fillers. Such utterances are referred to as novel if they appear to have been constructed through activation of an entrenched syntactic template and selection of lexical elements which cannot be traced to any other language produced.

Looking at child data, one wonders what determines the division of labour between the deployment of multiword frozen chunks, partially open frames, and completely open patterns (i.e., syntactic templates). One factor that has received much attention in the literature on language acquisition is the frequency of multiword combinations in the input: it appears that frozen chunks often found in child speech are also found with high frequency in child-directed speech. Cameron-Faulkner et al. (2003), for example, show such repetitiveness of child-directed speech in utterance initial positions of 12 English-speaking mothers with high correlation to the child's phrases built around these highly frequent items. In a follow-up study with languages with freer word order, such as Russian, German and English, Stoll et al. (2009) report that the input directed at two-year-old children is indeed lexically restricted, at least at the beginnings of utterances they study. High degrees of repetitiveness of items such as *That's a X* was found in the speech of all examined Russian-, German-

and English-speaking mothers. There were also intriguing differences, with English having the most and the longest frames, accounting for more of the input data than the two remaining languages, a finding explained by that language having the most restricted word order (Stoll et al. 2009).

Evidence from adult usage shows that productivity of a pattern does not necessitate the use of the completely empty pattern, which would be the equivalent of rule-based activation (Walsh et al. 2010). While one of the production routes is certainly a process which seems akin to a beads-on-a string assembly, this route is costly as it requires considerable cognitive work (Walsh et al. 2010). For example, when a child opens a sticker book, points at a missing sticker, and says *apple gone* or *house gone*, such cognitive work would be attributed to activating the rule that a subject noun precedes a participle. More likely, the child activates a slightly less schematic pattern: *gone* is preceded by a noun. Adult data indicate that the preferred route is to activate a construction via lexical means in a process referred to as unit-based recall (Walsh et al. 2010). This would be possible if the fixed element, *gone* in our example, first starts being used with one item that fills the slot more frequently than others, as would be expected of the versatile construction *it's gone*. As these words recur together in speech, they are expected to form a collocational bond that helps make their production more automatic and ensures smooth transition from one to the other during utterance (Walsh et al. 2010). In the process, the morphosyntactic relation between the elements is backgrounded, and they are recalled as one unit (Bybee 2001; Walsh et al. 2010). Importantly, this view assumes non-redundancy. The nature of linguistic representation allows for productive schemas to co-exist with less productive patterns and for constructions to be assembled via rule-based ('beads-on-a-string') and lexical ('unit-based') means, with, importantly, many gradations in between. This usage-based approach captures the dynamic nature of language in its continuity between grammar and lexis which are subject to constant change depending on one's individual experience of language usage (e.g., Bybee 2001).

*1.2. Our Research Questions*

Studies of monolingual children have provided ample evidence that children's journey towards adult-like competence originates in such 'slot-and-frame' schemas and that it is piecemeal and mostly lexically-based, at least in the early stages of acquisition. Usage-based studies of monolingual children reach back to Braine (1963) and his three-rule pivot grammar which sowed the first seeds for change in the way we now view early child language. More recent research into 'slot-and-frame' patterns has focused on verb-argument constructions (Keren-Portnoy 2006; Ninio 1999; Tomasello 1992), and interrogative constructions (Dąbrowska 2000; Dąbrowska and Lieven 2005). However, considering that most of the world is bilingual, there has been surprisingly little interest in how acquisition of slot-and-frame patterns proceeds in contexts where two languages are present in the environment. We aim to fill this gap by referring to data from a bilingual child exposed to Polish and English from birth.

The *first* research question we thus ask in our study is how the acquisition of such schemas proceeds longitudinally under conditions of bilingual exposure, from the first lexically fixed combinations produced to more open constructions. The idea behind the current article is that bilingual acquisition data may give us richer insight into how children build up their syntactic productivity. In the current study, we are particularly interested in the evidence for productivity given by CS in child data, since the child we will report on, like many other child study participants in the bilingual acquisition literature, grew up in a family in which there was very little CS in the input. Also, since input in the two languages is rarely equal, bilingual acquisition data show to what extent quantitative, and perhaps also qualitative, differences in the input in each language lead to differences in the acquisition of syntactic productivity. This would allow us to get a more sophisticated view of the role of frequency, including of its limitations rather than just the demonstration that it plays a pivotal role.

To the best of our knowledge, only one bilingual study of slot-and-frame patterns exists: Quick et al. (2018) report the constructing of language from both English and German input by a child called Tim recorded between the ages of 1;10–3;1. Using the 'Traceback' method (see below), the

study arrives at a quantitative breakdown of all bilingual constructions produced in four logically possible types of constructions.

(a)    Completely lexically fixed chunks, here referred to as frozen, e.g., *hilf-me* (help me): 18%

(b)    Creative combinations of multiple chunks, e.g., *let's kaputt-machen* (let's break it): 11%

(c)    Partially schematic constructions where the fixed element is either monolingual, e.g., *ich-kann-nicht X* (I can't X), or bilingual, e.g., *ich want X* (I want X): 60%

(d)    Other, e.g., utterances with no schemas, e.g., *ein open Mama* (one open Mama): 11%

If we combine the categories 'b' and 'c' as both instantiating partial schematicity (where 'a' is lexical and 'd' is syntactic as these terms are commonly understood), it is clear that most language production concerned partially schematic units, i.e., frames with open slots. We may expect that most of these bilingual constructions involve a frame in one language and a slot filler in the other. This invites our *second* research question about the extent to which productivity of a given pattern results in openness of that pattern to items from either language. We expect that partial productivity is only part of the explanation as some of the CS in category (c) occurs within the frozen constructional frames (Quick et al. 2018). If our expectation is confirmed, we will examine child usage data for any further evidence which could help us to explain why CS occurs at certain points in the constructions.

The very fact that CS occurs at all in Tim's data also needs explaining. The parents used solely English at home while German input was delivered in nursery. This suggests that frozen bilingual constructions must have resulted from the child's own language usage rather than from hearing parental CS. Usage-based linguistics tends to privilege the passive part (witness the emphasis on input), but of course usage is both input and own production. Without attention to the latter, usage-based linguistics runs the risk of appearing as a sophisticated update of behaviourism and its fascination with imitation. Partially schematic constructions are by definition sites of productive (or 'creative') language use and a gateway to more abstract syntax: the schema may be entrenched by the time it is used, but filling its open slot with a novel item not used in that slot before means a novel utterance has been produced. Another question which thus needs to be addressed is how the building of constructions resembles parental input and the child's own experience of language practice. More concretely: how come children codeswitch when the input emphasizes separation of the languages. This is our *third* research question, and by referring to evidence provided to address the first two questions, we will suggest that the answer has to do with the development of syntactic productivity and with the relative unnaturalness of language separation.

### 1.3. Our Contribution to Bilingual Research

Our study aims to use the slot-and-frame approach in relation to bilingual data to expand on what is already known about bilingual acquisition through studies produced to date. Such studies have been particularly helpful in highlighting the general trends observed in children studied across linguistic communities. It is now well established that CS is commonplace before the age of two but it tends to phase out if both languages are kept separate in the child's environment (Nicoladis and Genesee 1996; Redlinger and Park 1980; Volterra and Taeschner 1978; Paradis and Nicoladis 2007). Some of this early CS may be due to lexical gaps: around the age of two children sometimes use a word from another language because they do not have an appropriate translation equivalent (Nicoladis and Secco 2000; Quay 1995). However, with increasing proficiency in both languages, bilingual children learn to use more translation equivalents (Legacy et al. 2016) and this presumably allows them to figure out how to use them in context sensitive ways. As children as young as two display interlocutor sensitivity in that they adapt their speech to that of their caregivers (Deuchar and Quay 2000; Lanza 1997; Nicoladis and Genesee 1996), early CS also appears to be a function of parental language use: children codeswitch more if CS is not challenged by their parents (Lanza 1988 but see Deuchar and Muntz 2003; Nicoladis and Genesee 1998); they also codeswitch more when CS is modelled in the

input (Comeau et al. 2003). However, parental adherence to the OPOL strategy does not guarantee lack of mixing by children (Mishina-Mori 2011).

Of particular relevance to the qualitative nature of CS is also the observation that if the child's two languages display asymmetry in acquisition, with one language developing faster than the other, such asymmetry will determine the nature of words which are used in bilingual combinations. Dominance is likely to be of importance in mixing as most bilingual children are dominant in one of their two languages (Gathercole 2016; Paradis and Nicoladis 2007) and this shows in various measures, including amount of exposure to both languages (Unsworth 2015; Nicoladis et al. 2018), the MLUs in both languages (e.g., Quick et al. 2018), the number of TE equivalents available (Legacy et al. 2016; Nicoladis et al. 2018), parental reports and relative proportions of language used (Nicoladis et al. 2018). By referring to the Matrix Frame Model of CS (Myers-Scotton and Jake 2001) which assumes a strict division between grammar and lexis, it has been argued that it is usually the child's 'dominant' language which provides the functional frame while the language used less frequently provides individual content words (Bernardini and Schlyter 2004; Cantone 2007; Gawlitzek-Maiwald and Tracy 1996; Petersen 1988) though more recent studies show that frames can sometimes be derived from the weaker language (Müller et al. 2015). To demonstrate the relationship between dominance and mixing, Petersen (1988), for example, constructs the Dominant Language Hypothesis which allows her to define the dominant language as one which contains fewer mixes. Under this hypothesis, grammatical morphemes from the dominant language can occur with lexical items of either the dominant or the weaker language; however, grammatical morphemes from the weaker language can occur only with lexical morphemes from that language. Meanwhile, the accounts presented by Gawlitzek-Maiwald and Tracy (1996) and Bernardini and Schlyter (2004), for example, explain how CS proceeds when one language is dominant and provides a functional skeleton for the weaker language to grow into. However, it remains unclear whether it is dominance which exerts influence on how languages are mixed or the other way round. More importantly, circularity is a danger: we explain a particular asymmetry in the data with reference to dominance, but use the asymmetry to establish dominance. The concept becomes more useful, we will argue, if dominance is linked to which language provides more of the syntactic frames (including partially schematic constructions) that host slot fillers from the other language and especially if this asymmetry can be linked to differences in the child's linguistic experience and thus to higher degrees of entrenchment for that language's partially schematic units.

The three research questions introduced earlier in this section will be addressed by referring to data from Polish and English, a language pair not studied before for the acquisition of early constructions or CS. We find that the typological distance between Polish (a highly inflected language) and English (a fusional language) allows us to ascribe a language index more easily to individual words and patterns.

## 2. Materials and Methods

### 2.1. The Participant

This study is part of a project which examined one child's productions in light of her language usage patterns, using diary data and video recordings (for further details see Gaskins 2017). Informed consent for inclusion of the child in the study was gained from her mother before the study was launched. The study was conducted in accordance with the Declaration of Helsinki, and the protocol was approved by the School of Sciences, History and Philosophy Ethics Committee at the University of London (code 2012-09).

The main participant of this study is Sadie, a first-born and normally developing child who presents a case of bilingual first language acquisition (BFLA). Sadie was born and raised in England and she heard English at home from her father who did not know any Polish. Polish, on the other hand, was heard regularly only from her mother, the only speaker of that language in her immediate environment, whose command of English did not go unnoticed by the child. In addition, the parents

spoke English with each other at home. In her second year of life, Sadie attended an English nursery three days a week, 10 h a day, and spent the remaining two weekdays addressed in Polish by her mother, and weekends with both parents. In the summer, both at the end of her first and second year of life, she spent two weeks in Poland each time, fully immersed in Polish. Additionally, once every three months, she was visited by her maternal grandmother who stayed with her for two weeks at a time and addressed her only in Polish. When Sadie was at home, her parents conformed fairly consistently to the OPOL strategy (one-parent-one-language): Sadie's Polish-speaking mother used 8 types and 12 tokens of individual English words over the course of the ten Polish recordings (vs. 1626 types and 9485 tokens of Polish words) while her English-speaking father used 16 types and 24 tokens of individual Polish words across the ten English recordings (vs. 1119 types and 13,675 tokens of English words).

Sadie's language acquisition is asymmetrical, for at least four reasons. First, the diary data reveal that throughout her second year of life Sadie received roughly 65% of her linguistic input in English. Second, at the age of 2;02 Sadie's word stock was 74% English (292 words) and 26% Polish (103 words). Third, when recorded on video speaking to her father, she used mostly English with only 2% of the words Polish. When recorded speaking to her mother, however, she used on average 90% English and only 10% Polish word tokens, at comparable and relatively stable rates throughout the data collection period (see also Gaskins 2017). This shows that English was her dominant language of interaction regardless of the language addressed to her. Lastly, at 1;10.16 her MLU measured in monolingual English utterances was 1.63 and increased to 2.35 by 2;05.11 while at 1;10.20 her MLU measured in monolingual Polish utterances was 1.03 and dropped to 1 word per turn at 2;04.15.

## 2.2. The Data

Following the Language Diary Method (De Houwer and Bornstein 2003), a diary was kept to record Sadie's development between 0;10.10–2;3.22. The diary contained quantitative information on the amount of input she received in each language such as which language was addressed to her in each 30-minute segment of the day; it also listed any new words and multiword combinations she produced. The diary was updated as and when a new word or word combination was heard or when an existing combination was heard combined with a new word. Between the ages of 1;11.01 and 2;0.10 the diary was updated throughout the day every single day as Sadie's mother was off work. After 2;0.10, this updating happened throughout the day on 4 days a week when Sadie's mother stayed at home and in the evenings on the remaining 3 days when Sadie's mother worked during the day. No diary entries were made when Sadie's mother was at work. Up to the age of 1;10 all new language was recorded, including single words and multiword combinations, but between 1;10–2;2 priority was given to new combinations as the sheer amount of language Sadie produced meant that it was impossible to record it all. Despite limitations of diary data which could not capture every single instance of language use, access to the diary gave us a privileged insight into slot-and-frame patterns from a very young age and it allowed us to capture the very first instance of when words were combined together in speech.

As a second source of data, 30 half-hour video recordings (1;10.16–2;5.11) were transcribed, amounting to fifteen hours' worth of interactions. These recordings are representative of three sociolinguistic contexts: there are ten recordings with Sadie's father where she was addressed in English, ten with Sadie's mother where she was addressed in Polish and ten with both parents present where she heard both languages. However, seeing that regardless of the context Sadie always preferred to speak English, all the data were collapsed into one dataset. All the recordings were made at dinnertime, followed by playtime which often involved looking at books, matching up animal cards and playing with Lego. Video recorded data allowed us to capture the most frequently produced combinations. These were then verified against diary data: if there was no CS in an utterance within a given schema on video, we verified if this also held for the diary data; if diary contained conflicting information, schemas were then shifted to the category of bilingual combinations.

*2.3. Data Analysis*

All Sadie's monolingual and bilingual utterances, as recorded in the diary, were examined using what we call a 'diary Traceback method'. This method was adapted from that used to analyse densely sampled corpora of recorded speech and to trace constructions back to those recorded in prior videotaped interactions (Dąbrowska and Lieven 2005; Lieven et al. 2009; Quick et al. 2018). In our study, to verify whether piecemeal acquisition holds for our context of bilingual exposure, we first traced construction development throughout the diary recording period. The tracing consisted in following longitudinally all the earliest two-word and multiword constructions noted down in the diary to establish which of their elements were frames and which the words selected to fill the slots in those frames. Depending on the data available, all the word combinations recorded in the diary were subsequently divided into three groups: frozen, novel and partially schematic. If neither of the two (or more) elements of a combination were ever seen to be replaced with another in speech, as was the case with *Thank you*, the combination was deemed to be a frozen unit, as there was no evidence that it had been built using a productive frame. If a combination seemed to have been assembled following a more abstract schema, it was deemed to be novel, even though there were not sufficient data to ascertain schematicity beyond doubt. It is possible that such combinations had been picked up holistically, yet they corresponded closely with existing partially schematic units (e.g., *Red car* corresponded with *Naughty X* and *Silly X*) which gave us reasons to believe that had been assembled productively. Everything else belonged to the final category of partially schematic constructions, which we will focus on in this paper. We will illustrate the development of such constructions with the example of *No X*. The word no was first produced at 1;02.10. Its first occurrence within multiword combinations was *no potatoes*, at 1;6.03. After that, *no* was also recorded with *bed* (*No bed* at 1;7.07) and *more* (*No more* at 1;7.20) and therefore the word *no* could now be considered to have given rise to a frame *No X*. Following the same logic, the phrase *I don't want X* emerged from a frozen chunk which was initially heard once in the combination *want it yoghurt* (1;9.21), a non-target like construction with two objects. Once the word *cheese* was also produced in the same frame (at 1;10.22), the combination *Want it X* was recognised as a frame. Its non-target like character persisted, perhaps because the conjoined usage of the words *want* and *it* is frequent (in common expressions such as *I don't want it* and *Do you want it?*). Eventually, Sadie stopped using the word *it* in the above frame, leading to more target-like usage of the verb-object construction, and the disappearance of the productive schema *Want it X*. By now, the word *want* started being used repeatedly in the extended combinations *I don't want it* and *I don't want X*.

Since our video recording schedule typically involved only 3–4 recordings per month, our video recorded data were not sufficiently dense to lend themselves to exactly the same kind of analysis as that adopted by other researchers (Dąbrowska and Lieven 2005; Lieven et al. 2009; Quick et al. 2018). Instead, once the video recorded data had been transcribed on CHAT, they were traced back to the slot-and-frame patterns from the diary and then linked to them using FREQ and KWAL commands on CLAN. Altogether 6465 of Sadie's utterances were examined of which 1717 were multiword. Of these, 198 were bilingual. We defined bilingual constructions as any utterances which contained at least one word from Polish and one from English. To link constructions' productivity to switch placement, we further examined the four most frequently produced monolingual and the six most frequently produced bilingual partially schematic utterances, as the latter were by far more common in the child's data. These constructions were further analysed in terms of type/token ratios (TTR) of slot fillers in order to establish their productivity. The more types of words used within the slot, the higher the TTR ratio and the more productive the slot (Bybee 2001). The constructions were also traced back to the diary to examine their earliest usage patterns, specifically the form in which they emerged early in acquisition.

## 3. Results

### 3.1. Schematic and Specific Units

We first look at all word combinations in Sadie's diary data. At this point we do not distinguish between English and Polish, and lump all data together, including combinations that instantiate CS. The most frequently produced monolingual and bilingual combinations will be explored in detail in Sections 3.2.1 and 3.2.2.

The results discussed here demonstrate how word combination, and in particular frame formation, proceeded in Sadie's bilingual acquisition. Overall, 315 tokens of combinations produced by Sadie in English, Polish or combining both languages were found in the diary between 1;4.17 and 1;2.00. All these were divided into the three schematicity categories, depending on the evidence available to support their categorization (see Figure 1). The first group were frozen multiword units, because no other combinations were found with either of the words. They account for 4% (*n* = 13) of Sadie's 315 types of multiword combinations (See Appendix A). Among them were imitations of phrases heard on TV, from books and from parents (e.g., *Wait for me*!) social phrases (e.g., *Well done*!), linguistic routines (e.g., *What's that*?) and compound nouns (*Bath time*).

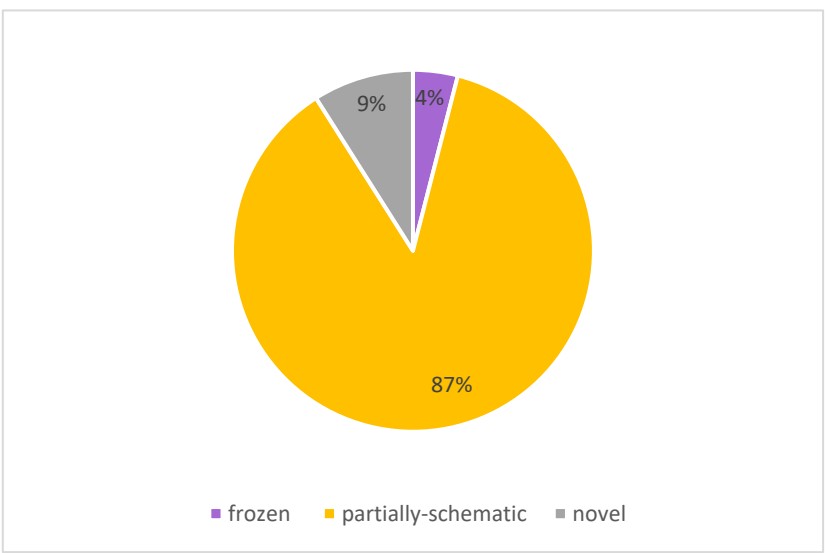

**Figure 1.** Schematicity of constructions in the diary data.

The second group of constructions, and those we will focus on in this paper, were partially schematic, with at least two examples for each lexically-based pattern. They included discourse routines (e.g., *Bye bye X*), questions (e.g., *Where's X*), noun phrases (e.g., *The X*), prepositional phrases (e.g., *In the X*), noun-based schemas (e.g., *Sadie X*), verb-based schemas (e.g., *Go away, X!*) and pronoun-based schemas (e.g., *Everybody X*) as well as vocatives (e.g., *Daddy, X!*). Some constructions were also included within others, e.g., *The X* is a construction in its own right but also part of *In the X* and *On the X*. Most fixed elements in the frames were functional items, such as social words, question words, adjectives, determiners, prepositions and functional verbs. Partially schematic combinations account for 87% (274) of Sadie's 315 multiword constructions; recall that the slot X can be either in English or in Polish (see Appendix B). The 67 types of partially schematic patterns evolved over the period of 9.5 months, with the first one emerging at 1;4.17 and the last just before the end of data sampling at 2;2.00. If we define the language of the frame as the language of the fixed element, 50 of the 67 patterns had English frames, 14 had Polish frames and three had a frame that fitted both languages. If we assume that partially schematic units reflect the emergence of syntax, this finding shows that under the conditions of imbalanced bilingual exposure Sadie experiences, the distribution of frames

mirrors her input: most frames come from English, which is Sadie's dominant language, with single words from English and sometimes Polish filling the open slot (marked as X in the examples above).

There was also a third group of 28 (9%) combinations that remained unclassified after identifying all frozen and partially schematic constructions and which were considered novel (see Appendix C). In these cases, no more than one example was found for a particular verb plus argument, which invited an interpretation that they had been constructed in accordance with a more abstract schema, i.e., both verb and argument were considered slot fillers in an entirely schematic unit. Given the limitations of the data, we cannot be sure of course whether there were no earlier occurrences of any of these words in these patterns: in other words, the Traceback method is a conservative method whose technical definition of novel combination likely makes us overestimate the proportion of novel combinations. Among them were noun-based schemas (e.g., *Red car*), Subject-Verb [SV] structures (e.g., *Baby's crying*), Verb-Subject [VS] structures (e.g., *Jedzie pociąg* 'is going the train'), Subject-Verb-Object [SVO] structures (e.g., *Ja chcę smoczek* 'I want the dummy'), imperatives (e.g., *Come back, pies*! 'come back, dog'), questions (e.g., *Has daddy got bicycle*?) and combinations of multiple schemas (e.g., *What happened, everybody*?). Most of the novel combinations were based around a verb. Investigating these instantiations is important, as they potentially show the emergence of syntax, i.e., the use of schemas more abstract than the constructions that are only partially schematic, which we focus on here.

All three groups make up the full constructional inventory of Sadie's output in the period under investigation, as far as our data allow its reconstruction, without regard for whether the constructions contained English or Polish lexical material. Most of her output was English. In the next section we will look at the division across the languages in more detail, focusing on the occurrence of CS. We will see that this mostly took the form of Polish lexemes used in English frames, not vice versa.

### 3.2. Evidence for Productivity vs. Motivation for CS in Schematic Units

It should not come as a surprise that Sadie codeswitches, despite the fact she was raised with the OPOL strategy and her parents did not codeswitch, which is confirmed by lack of mixed units among her frozen combinations. The method of tracing constructions forward in a diary of course is not perfect: it is always possible that a partially schematic construction instantiating CS was indeed heard in parental speech or, more likely, produced by Sadie but not recorded, but its conservative nature inspires confidence that Sadie's use of CS in such constructions illustrates her expanding grammatical competence. Children raised in OPOL surroundings are indeed routinely reported to go through at least a phase in which they mix their languages, suggesting it is a natural phenomenon (e.g., Mishina-Mori 2011).

The analysis in this section contributes to that literature, but we mainly want to explore the evidence for productivity that CS affords when found in the output of an OPOL-raised bilingual child. Technically speaking, inserting a foreign word should be possible for any partially schematic pattern. Examining which ones do in fact host foreign words should tell us something about what kinds of constructions attract CS and therefore play a role in accommodating loanwords. It may also tell us something about productivity, as constructions that host words of foreign origin may be the most productive patterns. To explore this in more depth, we zoom in onto the most frequently used constructions with high token numbers in the diary and on video, taking large numbers of slot fillers. In Sadie's data, there are two prototypes among these constructions in terms of their openness to CS: constructions whose instantiations are always monolingual and constructions that frequently have their slot filled by material from the other language. In this section, we analyse this difference and suggest an explanation. Constructions with low token numbers are not considered here: they may not have been captured in sufficient breadth to warrant meaningful analysis.

The asymmetry between the languages noted earlier also has implications for the CS in the data. Most monolingual constructions had fixed material only drawn from English (i.e., they are 'English constructions') and always hosted slot fillers drawn from English (Group 1). The few Polish schemas virtually always hosted slot fillers from the same language. The full inventory

of monolingual constructions from this group is presented in Appendix D Table A1. This set of constructions includes five of the ten Polish schemas, and 19 English constructions, including the 33 tokens of *The X* and 26 tokens of *I X*, the two most frequently produced monolingual partially schematic constructions. Appendix D Table A2, on the other hand, lists all instantiations of the 37 partially schematic constructions that did sometimes include CS (Group 2), e.g., all ten occurrences of *Bye-bye X*. Inspecting these data allows us to look for the patterns of productivity. Some patterns are less productive than others because they occur with only a limited set of complements (like personal names and vocative words such as *daddy*; such patterns might not be patterns in cognitively real terms but just collections of similar frozen units). The most productive patterns are the syntactically most interesting ones, and we study them with one overall question in mind–what accounts for the CS we see in some of these patterns but not the others? To follow from this, we advance an account for why the abovementioned extremely productive schemas *The X* and *I X* gave rise to solidly monolingual English instantiations.

### 3.2.1. Group 1: Mostly Monolingual Constructions

In the monolingual group (see Table 1) are the invariably monolingual *I X* and *The X* (the latter of which is also enclosed in longer monolingual constructions with prepositions such as *In the X* and *On the X*) as well as the nearly always monolingual *My X* and *Where's X* (for slot fillers see Appendix E). Note that all four have an English frame. Mind that in example 1, though some Polish words were slotted into these combinations, they occurred at points of the utterance other than after the pronoun central to the frame.

**Table 1.** Mostly monolingual constructions.

| Schema | Emergence | Diary Data | Video Data |
|---|---|---|---|
| *I X* | *I* emerged at 1;9.01 as part of *I can see you* and *I finished* | *I want cheese* (1;10.22); *I do it* (1;10.29); *I don't want it mleko 'milk' and I don't want it pies* 'dog' (1;11.04); *I don't want it ser* 'cheese' (1;11.05); *I don't want it spać* 'to sleep' (1;11.07); *I don't want it do domu* 'home' (1;11.12); *I go swim* (2;0.04); *I don't know* (2;0.08); *I want again* (2;0.18); *I need milk* (2;0.29); *I lost the dummy* (2;1.03); *I made it* (2;1.17) *and I did it* (2;1.18). | Overall, 55 tokens of the word *I* were recorded, always as part of constructions with 15 types of fillers used (TTR = 0.272). |
| *The X* | *The* emerged 1;08.07 as part of the phrase *On the floor* | *In the morning* (1;9.22), *Shut the door* (1;10.10), *Open the door* (1;10.13), *In the garden* (1;11.13); *I don't want it the bed* (1;11.26); *In the mouth* (1;11.29); *On the boat* (1;11.30); *Blow the candle* (2;0.00); *On the table* (2;0.10); *On the top* (2;0.15); *Wash the hands* (2;00.18); *Tickle the tummy* (2;0.19); *The light on* (2;0.20); *The bowl and Watch the beebies* (2;0.22); *On the window* (2;0.24); *In the bag and I lost the dummy* (2;1.03); *I lost the ball* (2;1.11) *and In the bathroom* (2;1.18) | Overall, 78 tokens of the word *the* were used, always as part of construction with 34 different types as fillers (TTR = 0.435). |
| *My X* | *My* emerged at 1;9.04 as part of the construction *My turn* | *My pencil* (2;1.27) | Used 66 times, always as a multiword unit, with 24 different types of fillers (TTR = 0.363). Two constructions *My X* contain slot fillers from Polish. |
| *Where('s) X* | *Where* emerged at 1;9.07 in the construction *Where are you* | *Where everybody* (2;0.05) and *Where mummy's slippers?* (2;0.11) | Used 61 times, always as a unit with 7 types of fillers (TTR = 0.114). |

### 3.2.2. Group 2: Frequently Bilingual Constructions

Next, we look at the constructions that were also highly frequent but often contained a slot filler from the other language (see Table 2): *More X, X gone, No X, Daj mi X* 'give me X', *(I don't) want (it) X, (Where) other (one) X*. One of these constructions is Polish. Overall, these six schema account for 51% tokens (*n* = 97) of mixed constructions recorded on video.

**Table 2.** Frequently bilingual constructions.

| Schema | Emergence | Diary Data | Video Data |
|---|---|---|---|
| *More X* | *More* emerged at 1;4.08, used holophrastically until produced in *More other one* (1;09.25) | *More woda 'water'* (1;11.08); *More pić* 'to drink' (1;11.13); *More kukurydza* 'corn' (1;11.14); *More woda* 'water' (1;11.15); *More kaczki* 'ducks' (1;11.24); *More tissue* (2;0.06); *I want more bread* (2;0.21) and *I want more ice-cream* (2;1.11) | Of 94 tokens of the word *more*, 28 were not followed by slot fillers and seven were followed by the word please. In the 53 used in constructions, there were 33 different types of fillers (TTR = 0.622). |
| *No X* | *No* emerged at 1;2.10 as a sole word and was used as such until 1;7.07 when recorded in *No bed* | *No more* (1;7.20); *No watch Piggle* (1;10.05); *No tickle, no want it* (1;10.25); *No woda* 'water' (1;11.13); *No ride sheep* (1;11.17); *No hiding* (1;11.17); *No juice* (1;11.25); *No eating* (1;11.30); *Got no boots, got no shoes* (1;11.30); *No ząbki* ' teeth' (2;0.01); *No want potty* (2;0.20); *No butys* 'shoes' (2;0.27); *No tak* 'like this' (2;1.10) and *No yours, Sadie's* (2;1.19) | Overall, 129 tokens of the word *no* were produced by Sadie on video of which 15 are not followed by a slot filler. The remaining 114 are produced with 69 different types of fillers (TTR = 0.438). |
| *X gone* | *Gone* emerged at 1;7.24 on its own and was used as a holophrase until 1;08.15 when recorded in *All gone* | *Tata gone* 'Daddy gone' (1;9.17); *Baby's gone* (1;10.21); *Hau gone* 'Woof gone' (1;11.02); *Pies gone* 'Dog gone' (1;11.09); *Everybody gone* (2;0.06) and *Reading gone* (2;0.19) | Overall, 8 tokens of the word *gone* were recorded on video, all in constructions with 7 different types of fillers (TTR = 0.875). |
| *Daj (mi) X* 'Give me X' | *Daj* 'give' emerged at 1;2.25; *mi* 'me' at 1;4.06. *Daj mi* 'give me' was first used at 1;4.17 | *Daj mi pots* 'give me pots' (1;4.22); *Daj mi that* (2;1.12) and *Daj mi keys* (2;1.13) | Overall, 23 tokens of *Daj (mi)* were recorded, of which two are produced without a slot filler. In the other 21, 5 types of fillers are used (TTR = 0.238). |
| *(I don't) want (it) X* | *Want it* emerged as an unprocessed chunk at 1;9.11 and was used on its own until 1;11.04 when used in a mixed construction *I don't want it pies* 'dog' | *I don't want it mleko* 'milk' and then at 1;11.05 *I don't want it ser* 'cheese' | Overall, 84 tokens of *want (it)* were recorded of which 42 are produced without a slot filler and 42 are used with 34 types of fillers (TTR = 0.809). |
| *(Where) other (one) X* | *Other one* emerged as an unprocessed chunk /auauan/ at 1;9.20, used on its own until 1;11.06 when recorded in *Other one peppa pig* | *Other one teddy* (2;0.29)–its use in constructions alternated with its use on its own. The frame was later extended through the addition of the word *where* to form a longer frame *Where other one* first recorded in a construction *Where other one cat* (2;1.06) | Overall, 79 tokens of *(where) other (one)* were recorded on video, of which six were produced without a filler and 73 with 33 types of fillers (TTR = 0.452). Typically produced in the context of a card game in which she was expected to find matching animals. |

The usage data show that generally constructions that contain CS some of the time (mean TTR = 0.581) were more productive than the ones that never do (mean TTR = 0.307). However, the data concerning productivity, at least the kind defined as a function of type frequency, are inconsistent. *The X* (TTR = 0.435) from Group 1 was as productive as both *(Where) other (one) X* (TTR = 0.452) and *No X* (TTR = 0.438) from Group 2. Likewise, the productivity of *I X* (TTR = 0.272), *My X* (0.363) and *Where('s) X* (TTR = 0.114) from Group 1 was comparable to that of the Polish construction *Daj mi X* (TTR = 0.238) from Group 2. This suggests that reasons for openness of constructions to CS need to be examined in more detail. A closer look at the usage patterns reveals that in Group 1, the words *I, the, my* and *where* all emerged as, and all remained parts of, constructions. Although a range of words

filled the slots in these constructions, thus potentially leading to segmentation of their component parts and increased productivity of the schema, the words of the frames were never used on their own, so it is not clear whether these words were conceptualized as individual linguistic entities. On the other hand, in Group 2 all the frames emerged first as individual words or longer multiword units and possibly became entrenched and conceptualized as such through holophrastic use. In the next section, we will summarize these findings and discuss their implications for theories of language acquisition, particularly in bilingual settings.

## 4. Discussion

We have demonstrated how the acquisition of constructions proceeds in a two-year-old exposed to Polish and English from birth, and who shows a preference for using the latter regardless of who her interlocutor is. Lumping output in both languages together, data produced by the diary generated 315 constructions recorded between 1;4.17 and 1;2.00 of which 4% (*n* = 13) were frozen, 87% (*n* = 274) partially schematic and 9% (*n* = 28) potentially novel. From among the 67 types of Sadie's partially schematic constructions, 50 had English and 14 Polish frames, while the three remaining frames could be interpreted as either Polish or English, a division that corresponds with the asymmetry in the input Sadie received from her environment. These findings allow us to answer our first question in that they confirm that much of the creativity of children's language use seems to be located in the use of slot-and-frame patterns (e.g., Keren-Portnoy 2006; Lieven et al. 1992; Quick et al. 2018), and to consist of the filling of slots with new lexical items.

To address the second question about the openness of partially schematic constructions to CS, we examined a pool of the most frequent monolingual and bilingual units from the video recordings and supplemented them with the diary data. Given the child's dominance in English, this mostly meant a comparison of constructions mostly or only instantiated as fully English chunks (Group 1) and English constructions that often contained some Polish material, usually a content word (Group 2). The usage data show that constructions that never contain CS are less productive (mean TTR = 0.307) than those that contain CS some of the time (mean TTR = 0.581). This suggests that the occurrence of CS in a construction is a sign of its productivity, or that the usefulness of a construction in hosting new words, including words from the other language, is what drives its productivity. However, the TTRs of individual constructions within each group are dispersed across a wide range, which suggests that type frequencies of slot fillers may not be the most accurate way of predicting the openness of such constructions to material from the 'other' language. Our data suggest rather that this openness has something to do with the usage patterns of a frame from early emergence through to subsequent use. Examination of the frames in Group 1 suggests that their production always came about by virtue of being part of longer stretches of speech. This lack of articulatory and semantic autonomy may explain why they were never or rarely combined with Polish items. On the other hand, examination of frames from Group 2 shows that they were not tightly attached to other words, and therefore must have gained some articulatory autonomy, allowing them to be combined more freely with items from both languages. This shows that in a context where CS is rarely or never modelled in the input, productive assembly of bilingual speech is facilitated by the words having more independent semantic identity and therefore having been entrenched through solitary use.

Other factors which contribute to CS most likely include whether or not the slot projects for elements typically amenable to CS, such as semantically specific words. All schemas from Group 2 project for a noun, almost any noun. Taking *More X* as an example, more and X are relatively autonomous, in the sense that they both contribute semantics that is essential for the meaning of the whole and both *more* and *X* (whatever it is that fills the slot) will often occur without the other. For schemas from Group 1, such as *I X* or *The X*, that is not the case: *I* and *the* are more dependent on co-occurring material than *more* is. We suggest this is why we find *more* on its own and not *I* or *the*. Due to this difference in autonomy and dependency *I* and *the* virtually always trigger further material with which they form multiword units in Sadie's mind, and by virtue of the relatively monolingual modes

she is usually in, all or most of these units will be completely English. Furthermore, in *I X* (though in no other constructions from Group 1) the *X* category actually consists of many multiword chunks with different structural properties, so that in some sense it is less productive. This, ultimately, has to do with the distribution of semantic autonomy and dependence, and may explain why 'productivity' by itself, at least the kind understood merely as a product of type frequency, is insufficient as an explanation.

Lack of CS in *The X* could also be linked to its sheer frequency in speech: as it does not have competitors in other definite articles, apart from those nouns which require zero article, it is in wide use and initially children may not be aware that *the* is a separate word. This may sound counterintuitive in light of what we know about type frequency. After all, if the recurs with a high number of nouns in the slot of the construction, we should expect high productivity of that slot (Bybee 2001) and, by extension, high levels of CS within that slot. However, *The X* is virtually impervious to CS. It may be useful to refer to some cross-linguistic data to help us to explain this observation. French deploys a range of definite articles, depending on gender (l' as in *l'amour* 'love'; la as in *la vie* 'life'; le as in *le matin* 'the morning' and les as in *les bonbons* 'candies'). In the early acquisition of *Definite article X*, whole noun phrases are replicated as whole constructions and no errors are evident in use (Leroy-Collombel 2010). Once the concrete constructions have been analysed, gender errors begin to occur (e.g., *le poule* 'the hen' instead of *la poule*) and finally children start to use the relevant determiners with the right gender (Leroy-Collombel 2010). The case of French thus shows initial tight attachment of particular articles to particular nouns which is likely the result of rote-learning. As French children hear contrastive use of three different articles, they learn that the articles are separate elements. By extension, we speculate that if the English *the* had competitors in other English definite articles, children would be forced to experiment with its use earlier on and they would figure out sooner how to use it productively. We suspect that this typologically determined ease of detachment of articles from nouns has some implications for CS. Let us move on to the example of German, a language with three definite articles (der as in *der Mann* 'the man'; die as in *die Frau* 'the woman' and das as in *das Brot* 'the bread'). Quick et al. (2018) discuss the bilingual acquisition of one child, Tim, and report that he switches within noun phrases when the definite article is German, e.g., *Und das X* and *And die X*, but not when it is English, i.e., *The X*. Presumably, high type frequency of definite articles in German leads to quicker emergence of the determiner category which, in turn, facilitates CS. By extension, the fact that there is only one definite article in English leads to a delayed emergence of this category. In the case of German *Article X* construction, it is thus the bilateral processing of that construction which facilitates CS within it because it triggers schematizing at an earlier stage. Our data confirm that the unilateral processing of *The X* in English leads to a slower emergence of partial schematicity and therefore it does not trigger CS within the construction in our data.

These findings have important implications for our understanding of the relationship between productivity and CS. One main observation was that in Sadie's data partial schematicity accounts for CS only to some extent. Is this because overall productivity is insufficient in explaining CS or because partial schematicity is not a sufficient determinant of CS? We show that most of the child's productions can be classed as partially schematic, regardless of whether they are bilingual or monolingual, and indeed that some of such partially schematic constructions remain impervious to CS. We also show that the words which facilitate CS are those which have been entrenched through autonomous use; and that some lexically fixed frames may require bilateral processing of the whole construction to trigger CS, as in the case of *The X*. This invites our conclusion that type frequency leading to partial processing of a schema is not a sufficient predictor of CS: some constructions may need more than just partial productivity though this would need to be confirmed in future research on a larger set of bilingual constructions.

In answer to the third question asked in this study, as to why Sadie codeswitches when her input emphasizes the separation of her languages, we suggest that CS is simply a reflection of her emerging syntactic productivity. As most of Sadie's frames are English, she switches to English in order to be able to say more regardless of the language of interaction. Her CS is likely supported by many factors

other than the way in which the two languages are presented in the input. One of them is higher entrenchment some words compared to their translation equivalents which facilitates their access and retrieval (see Quick et al. forthcoming). For example, most of Sadie's frames are English and the fact that they are activated even when Sadie is addressed in Polish suggests they must be more entrenched and easier to access. Sadie's CS thus shows limitations of input in accounting for language use: despite being raised in an OPOL environment, she still combined her two emerging languages together in speech. Additionally, purely on the basis of the description of the family linguistic situation one could have expected that Sadie's Polish would be very rudimentary. However, high numbers of CS utterances as well as some novel Polish combinations show that extensive input is perhaps not needed to build up some decent degree of competence and self-confidence in the minority language, an issue often discussed in the literature on Family Language Policy. The observation of CS also shows that Sadie, who does not experience intraclausal CS in the input, is really 'working' her languages. Particular frames are especially productive in this way; and we can see how the typical characteristics of insertion CS (i.e., grammatical frame from one language hosts content words from the other) could develop if Sadie would continue to produce mixed speech. Whether she does or not is mostly dependent on sociolinguistic factors.

Finally, Sadie's usage data also show language dominance to be just a by-product of more basic processes of usage-based selection of words and constructions combined with sociolinguistic pressures on a child that stimulate an awareness of the language affiliation of these words and constructions. That language affiliation comes from two sources: the natural abstraction of knowledge from co-occurrence patterns, which holds for all humans everywhere, and sociolinguistic emphasis on language separation, which may be strong (the usual case in the bilingual acquisition literature) or not (rarer, perhaps because of empirical bias or perhaps because of social reality). Clearly, naturalistic data can only be indicative due to production limitations and the restricted context of recordings. Therefore, ideally future studies should complement our knowledge by investigating the questions we asked here under experimental conditions.

## 5. Conclusions

In this study, we have shown that like in monolingual development, bilingual constructions can be also accounted for by the emerging slot-and-frame patterns. Access to frequently produced monolingual and bilingual constructions allowed us to highlight a limited role of type frequency for the processing of words in speech. Their autonomous use as well as bilateral processing of constructions, which they are part of, appears to be a better predictor of their productivity and their readiness to enter in combination with words from another language. Despite showing links between the child's own language usage and the patterns observed in her CS, we have also highlighted limitations of input in predicting a child's own language outcomes. The case of the child we studied shows that despite being raised in an OPOL context, she went through at least a phase of combining words from her two languages which suggests that CS is a natural manifestation of the bilingualism that results from being raised with two languages.

**Author Contributions:** Conceptualization, D.G., A.B. and A.E.Q.; methodology, D.G.; formal analysis, D.G. and A.B.; data curation, D.G.; writing—original draft preparation, D.G.; writing—review and editing, D.G., A.B. and A.E.Q.; visualization, D.G.

**Funding:** This research received no external funding.

**Acknowledgments:** Our sincere thanks to the editors of this issue as well as the two anonymous referees, all of whose comments helped to improve the paper. All the remaining weaknesses are our own.

**Conflicts of Interest:** The authors declare no conflicts of interest.

## Appendix A

Frozen constructions

(a) Imitations of phrases heard on TV and from books: *Are you ready to go? Don't worry my friend; What does she look like?*

(b) Imitations of phrases most likely heard from parents: *Pies cicho!* 'Dog; quiet!'; *Do domu pies!* 'Home dog!' *Oh there it is; Wait for me!; What have you got?*

(c) Compound nouns: *Bath time*

(d) Social phrases: *Well done; Night night*

(e) Linguistic routines: *Co to?* 'What's that?'; *W tę stronę* 'This way'

**Appendix B**

Partially schematic units

(a) Discourse routines: *Bye-bye X; Papa X* 'bye bye X'; *Hello X; Hi X; More X; Jeszcze X* 'more X'; *No X; Nie X* 'Not X' *Yes X; Thank you X; X gone; Nie ma X* 'not there X'; *X please*

(b) Questions: *What's that X?; Where/where's X?; Where other one X?; X [where] are you? What X?*

(c) Noun phrases: *The X; My X; Mine X; Moje X* 'my X'; *This X; Two X; X's turn; Other one X; Naughty X; Silly X*; possessives (*Sadie X* = English and Polish word order; *X Sadie*–Polish word order)

(d) Prepositional phrases: *X on; In the X; On the X; Do X* 'in the direction of X'

(e) Verb-based schemas: imperative (*Tickle X; Watch (it) X; X come!; Daj mi X!* 'give me X'; *X look!; Idź X!* 'Go away X!'); affirmative (*X coming; X śpi* 'is asleep'; *I'm gonna X; Jedziemy do X* 'we're going to X'); requests (*I want [it] X; No want X; I don't want it X; I lost the X; I need X*); demonstrative (*To jest X* 'this is X'; *Got no X; Jest X* 'is X'; *Look X; X's here*)

(f) Noun-based schemas: with verbs missing: *Sadie X* (e.g., *Sadie bicycle too; Sadie in there*); with verbs included: *Sadie X* (e.g., *Sadie clean up; Sadie otworzy* 'Sadie will open'; *Sadie broke it; Sadie wants bicycle; Sadie jest tutaj* 'Sadie is here') and *X the door* (*Shut the door! Open the door!*)

(g) Pronoun-based schemas: *I X* (e.g., *I see you; I made it; I did it; I do it; I go swim; I finished*) and *I'm X-ing* (e.g., *I'm swimming; I'm cleaning; I'm bouncing a ball; I'm coming to get you*); *X you* (e.g., *Thank you; Bless you*) and *Everybody X* (with verbs missing: *Everybody shower; Everybody up; Everybody bicycle; Everybody apple; Everybody tired; Everybody ząbki* 'Everybody teeth') and *Everybody X* (with verbs included: *Everybody sit down!*)

(h) Vocatives: *Daddy X!* (e.g., *Daddy have a go! Daddy help Sadie! Daddy śpij!* 'Daddy sleep!'); *Tata X!* 'Daddy X!' (e.g., *Tata stop!* 'Daddy stop!'; *Tata come!* 'Daddy come!'; *Tata come back!* 'Daddy come back!'); *X tata!* (*Łap tata!* 'Catch daddy!'; *Patrz tata!* 'Look daddy!'; *Idź tata!* 'Go away daddy!'); *X daddy* (*Stop it daddy!; What you doing daddy?*); *Mummy X* (e.g., *Mummy go swim! Mummy hiding? Mum why going?; Mummy what you doing? Mummy daj!* 'Mummy give!'; *Mummy why going?*)

**Appendix C**

Schematic constructions

(a) Noun-based schemas: *Jeden new book* 'One new book'; *One more; Red car; Ładna bluzka* 'A nice top'

(b) SV structures: *Baby's crying; Świnka Peppa hiding* 'Peppa pig is hiding'; *Dzidzia płacze* 'Baby's crying'

(c) VS structures: *Boli pupa* 'is hurting the bum'; *Jedzie pociąg* 'is going the train'

(d) Complete SVO structures: *Ja chcę smoczek* 'I want the dummy'; *Daddy take it* (meaning 'daddy took it') and SVO with the object missing: *We're making; Mummy washing*

(e) Imperatives: *Tortoise eat!; Come back pies!* 'Come back dog'; *Stay there! Siadaj tutaj* 'Sit here'; *Idź do domu* 'Go home'; *Tutaj sok!* 'Here juice'; *Blow the candle! Get it Lego! Zdejmuj buta* 'Take off the shoe' *Help me! Wash hands! You get it!*

(f) Questions: *Has daddy got bicycle?*

(g) Combinations of multiple schemas: *What happened everybody? Bless you mummy!*

## Appendix D

**Table A1.** All the types of monolingual partially schematic constructions (NB. Constructions *Sadie X* and *X Sadie* are excluded from analysis as they are language neutral).

|  | X is in the same language as the frame (examples from diary—left vs. Video-right) | |
|---|---|---|
| Hello X | *Hello moon*<br>*Hello everybody* | *Hello Maisy*<br>*Hello moon*<br>*Hello bear* |
| Jeszcze X 'more X' | *Jeszcze sok* 'more juice'<br>*Jeszcze zrobic* 'more to do' | *Jeszcze kredę* 'more chalk' |
| Nie ma X 'not there X' | *Nie ma smoczek* 'dummy not there'<br>*Nie ma dzidzi* 'baby not there'<br>*Nie ma taty* 'daddy not there' | 0 |
| What's that X? | *What's that noise*<br>*What's that poo*<br>*What's that button* | *What's that noise* |
| What X? | *What happened?*<br>*What are you doing?* | *What you doing the toilet*<br>*What else shall we draw*<br>*What am I gonna do* |
| The X | *The bowl; The ball; The door; The dummy; The light; The garden; The mouth; The bag; The top; The window; The boat; The table* | *The juice; The ball; The park; The garden; The flower; The trousers; The house; The beep beep; The toilet; The baby; The lights; The train; The sheep; The drum; The drums; The shakers; The belly; The moon; The tickets; The train; The ding dong* |
| This X | *This way*<br>*This one* | *This one; This way; This one cat; This one elephant* |
| Silly X | *Silly Pinkie* | 0 |
| X on | *Shoes on*<br>*Hat on*<br>*The light on*<br>*Trousers back on* | 0 |
| In the X | *In the garden mummy!*<br>*In the mouth*<br>*In the bag*<br>*In the bathroom*<br>*In the morning* | *In the garden* |
| On the X | *On the boat*<br>*On the table*<br>*On the top*<br>*On the window* | *On the toilet; On the train; On the belly* |
| Do X 'in the direction of X' | *Do domu* 'to the house'<br>*Do babci* 'to grandma's'<br>*Idziemy do park* 'we are going to the park' | *Do domu* 'to the house' |
| Tickle X | *Tickle the tummy*<br>*Tickle me* | *Tickle daddy; Tickle mummy* |

**Table A1.** *Cont.*

| Watch (it) X | *Watch the Beebies*<br>*Watch it Peppa Pig* | 0 |
|---|---|---|
| Idź X 'go away X' | *Idź do do domu pies* 'go home dog!'<br>*Idź osa!* 'go wasp' | 0 |
| Jedziemy do X 'we are going to X' | *Jedziemy do park* 'we are going to the park'<br>*Jedziemy do babci i dziadka* 'we are going to grandma's and grandpa's' | 0 |
| No want X | *No want it*<br>*No want potty* | *No want it* |
| I lost the X | *I lost dummy*<br>*I lost the dummy*<br>*I lost ball* | *Lost my baby; Lost the mummy; Lost your mummy* |
| Got no X | *Got no boots*<br>*Got no shoes* | 0 |
| X's here | *Mummy's here* | 0 |
| X the door | *Open the door*<br>*Shut the door* | *Open the door please* |
| I X | *I see you; I made it; I did it; I go swim; I finished* | *I got playdough; I found this; I drop; I want (it) X; I don't want (it) X; I got a cereal; I finished; I don't know; I say; I gonna drawing too; I find zigzag; I not finished; I got cooking; I have one; I need my torch; I play the drums; I need the shakers; I swim; I gonna do; I like; I play X* |
| I'm X-ing | *I'm swimming; I'm cleaning; I'm bouncing a ball; I'm coming to get you* | *I'm drinking the juice; I'm drawing; I'm drawing elephant; I'm taking my ( … )* |
| X; daddy | *Stop it daddy!; What you doing daddy?* | *No daddy; Thank you daddy; Very good daddy; Well done daddy; What's wrong daddy; Yes please daddy; Help daddy; Hat daddy; Hi daddy; Help me daddy; There you go daddy; You play this daddy; Yeah daddy; Look daddy; Yoghurt daddy; Come on daddy; More please daddy; Wake up daddy* |

**Table A2.** All the types of bilingual partially schematic constructions.

| | X is in the same language as the frame (examples from diary—left vs. video-right) | | X is NOT in the language of the frame (examples from diary—left vs. video-right) | |
|---|---|---|---|---|
| Nie X 'not X' | *Nie ma* 'not there'<br>*Nie do domu* 'not home'<br>*Nie pies* 'not dog'<br>*Nie kot* 'not cat' | *Nie come* 'not come' | *Nie jogurt* 'not yoghurt'<br>*Nie idź* 'don't go'<br>*Nie dobre* 'not nice'<br>*Nie wolno* 'not allowed'<br>*Nie to* 'not this' | *Nie garden* 'not garden' |

**Table A2.** *Cont.*

| | | | | |
|---|---|---|---|---|
| X please | *More please*<br>*Yes please*<br>*Here please* | *Yes please*<br>*No hand please*<br>*That one mummy please*<br>*Daddy please*<br>*Leave that please*<br>*More please*<br>*Help Sadie please*<br>*Tooh tooh please*<br>*Milk please*<br>*Baby please*<br>*Ticket please*<br>*Open the door please*<br>*My flowers please*<br>*More please*<br>*Okay please*<br>*Colour please*<br>*I play the ding dong the drum please*<br>*I'm play the drum please* | 0 | *Rysować please* 'to draw please'<br>*Daj mi that* please 'give me that please'<br>*Loda please* 'ice-cream please' |
| Where('s) X? | *Where's daddy?*<br>*Where's mummy go go?*<br>*Where everybody?*<br>*Where mummy slippers?* | *Where's daddy*<br>*Where's my mummy*<br>*Where's my torch*<br>*Where's the other part*<br>*Where's more flowers*<br>*Where's the one*<br>*Where's my elephant*<br>*Where are you*<br>*Where's my everybody*<br>*Where's my cut cut cut*<br>*Where's more rice* | 0 | *Where's my koń* (horse) |
| My X | *My pencil*<br>*My turn* | 0 | *My mummy; My turn; My teddy; My spoon; My daddy; My book; My hat; My mummy's book; My juice; My trousers; My torch; My teeth; My raisins; My water; Oh my gosh; My tickets; My everybodies; My cut cut cut; My jam* | *My koń* 'my horse' |
| Two X | *Two ball*<br>*Two doll* | *Two word; Two words; Two cats; Two stickers; Two cake;* | 0 | *Two zdjecia+s* 'two photos+s' |
| Other one X | *Other one Peppa Pig*<br>*Other one teddy* | *Other one tattoo; Other one ding ding; Other one cat; Other one cake; Other one ding dong; Other one butterflies; Other one pig; Other one stickers; Other one tiger; Other one fork; Other one spoon* | 0 | *Other one kot* 'cat' |
| Naughty X | *Naughty daddy* | 0 | 0 | *Naughty świnka* 'piggy' |
| To jest X 'this is X' | *To jest pies* 'dog'<br>*To jest kot* 'cat' | *To jest świnka* 'pig'<br>*To jest truskawkis* 'strawberries' | 0 | *To jest fish*<br>*To jest banana*<br>*To jest horse*<br>*To jest sheep* |
| Jest X 'here is X' | *Jest tramwaj* 'here is the tram'<br>*Jest tata* 'here's daddy'<br>*Jest woda* 'here's water'<br>*Jest tutaj* 'is here' (tutaj = here) | *Jest owca* 'here is sheep'<br>*Jest dobre* 'is tasty' | 0 | *Jest more* 'here is more'<br>*Jest Sadie's* 'is Sadie's'<br>*Jest sandwich* 'here is a snadwich' |

**Table A2.** *Cont.*

| | | | | |
|---|---|---|---|---|
| X; tata! 'X; daddy!' | *Łap tata!* 'Catch daddy!'; *Patrz tata!* 'Look daddy!'; *Idź tata!* 'Go away daddy!' *Jeszcze tata* 'more daddy'; *Daj tata* 'give daddy' | 0 | 0 | *Well done tata; No want tata; No tata; What you doing tata; Oh dear tata; Oh no tata; More tata; Come on tata* |
| Bye-bye X | 4 *Bye bye daddy Bye bye toys Bye bye water Bye bye Maker* (Mister Maker from TV) | 3 *Bye bye tattoo Bye bye food* | 2 *Bye bye kaczka* 'duck' *Bye bye tata* 'daddy' | 1 *Bye bye mama* 'mummy' |
| Hi X | 0 | *Hi daddy* | *Hi kaczka* 'duck' *Hi mleko* 'milk' | 0 |
| Papa X 'Bye-bye X' | 3 *Papa kaczka* 'bye bye duck' *Papa wieloryb* 'bye bye whale' *Papa woda* 'bye bye water' | 0 | 1 *Papa baby* 'bye bye baby' | 0 |
| Thank you X | *Thank you train* | *Thank you daddy Thank you mummy* | *Thank you tata* 'daddy' | 0 |
| More X | 4 *More berriesMore cakeMore breadMore other one* | 22 *More juice More caterpillar More painting More milk More cereal More fruit More banana More car More pasta More monkey More monkeys More lion More sheep More flowers More book More cheese More please More yoghurt More cake More what More rice More bread* | 4 *More woda* 'more water' *More pić* 'more to drink' *More kukurydza* 'more corn' *More kaczki* 'more ducks' | 14 *More mleko* 'more milk' *More jeszczes* 'more mores' *More jedens* 'more ones' *More jeden* 'more one' *More kredę* 'more chalk' *More jeszcze kredę* 'more more chalk' *More małpa* 'monkey' *More owca* 'sheep' *More pies* 'dog' *More kura* 'hen' *More other one kura* 'hen' *More owcas* 'sheep+s' *More modelina* 'playdough' *More, tata* 'more, daddy' |

**Table A2.** *Cont.*

| | | | | |
|---|---|---|---|---|
| No X | 12<br>*No potatoes*<br>*No bed*<br>*No more*<br>*No rabbit*<br>*No bath*<br>*No hair brush*<br>*No watch Piggle*<br>*No tickle*<br>*No want it*<br>*No ride sheep*<br>*No hiding*<br>*No that one* | 47<br>*No airplane*<br>*No green apple*<br>*No; mummy*<br>*No potato*<br>*No sausage*<br>*No breakfast*<br>*No juice*<br>*No juice in here*<br>*No drinking the juice*<br>*No one piece*<br>*No play the park*<br>*No more juice*<br>*No hand please*<br>*No my turn*<br>*No big cat*<br>*No banging*<br>*No bath*<br>*No fingers*<br>*No lion*<br>*No hat*<br>*No wake up*<br>*No fish*<br>*No potatoes*<br>*No wipe it*<br>*No wipe it me*<br>*No ready*<br>*No put up*<br>*No me*<br>*No mine*<br>*No it's me*<br>*No take our pencil*<br>*No helpYou no help*<br>*No yellow*<br>*No more*<br>*No Peppa Pig*<br>*No eating*<br>*No cheese*<br>*No sausage*<br>*No like*<br>*No seven*<br>*No Lailani birthday*<br>*No Sadie birthday*<br>*No drawing*<br>*No snail*<br>*No careful*<br>*No eat it* | 3<br>*No cześć* 'no hello'<br>*No mleko* 'no milk'<br>*No butys* 'no shoes' | 18<br>*No tata* 'no daddy'<br>*No want tata* 'daddy'<br>*No jogurt* 'yoghurt'<br>*No kawalek* 'piece'<br>*No rączki* 'hands'<br>*No świnka* 'piggy'<br>*No krowa* 'cow'<br>*No pies* 'dog'<br>*No ziemniaczki* 'potatoes'<br>*No tutaj* 'here'<br>*No chcę* 'I want'<br>*No miś* 'teddy'<br>*No jedz* 'eat!'<br>*No chleba* 'bread'<br>*No kąpać* 'to bathe'<br>*No sok* 'juice'<br>*No truskawkis* 'strawberries'<br>*No dziękuję* 'thank you' |
| Yes X | 1<br>*Yes please* | 6<br>*Yes please*<br>*Yes trolley*<br>*Yes song Peppa Pig*<br>*Yes danceYes car*<br>*Yes flowers* | 1<br>*Yes do domu* 'yes home' | 3<br>*Yes mleko* 'milk'<br>*Yes słoń* 'elephant'<br>*Yes samolot* 'plane' |
| X gone | 4<br>*All gone*<br>*Baby gone*<br>*Everybody gone*<br>*Reading gone* | 6<br>*Butterfly gone*<br>*All gone*<br>*Pasta gone*<br>*Mama* 'mummy' *gone*<br>*Ice-cream goneBaby's gone* | 2<br>*Tata gone* 'daddy gone'<br>*Pies gone* 'dog gone' | 3<br>*Dzidzia gone* 'baby gone' |

**Table A2.** *Cont.*

| | | | | |
|---|---|---|---|---|
| Where other (one) X | 0 | 11<br>*Where other one cat*<br>*Where other one monkey*<br>*Where other one fish*<br>*Where other one sheep*<br>*Where other one zebra*<br>*Where other one kangaroos*<br>*Where other hopping*<br>*Where other butterfly*<br>*Where other penguin*<br>*Where other one butterflies*<br>*Where other one fork* | 0 | 4<br>*Where other one pies* 'dog'<br>*Where other one zwierzęta* 'animals'<br>*Where other one owca* 'sheep'<br>*Where other one owcas* 'sheep+s'<br>*Where other one misiu* 'teddy'<br>*Where other one auto* 'car'<br>*Where other one kaczkis* 'ducks+s' |
| X (where) are you | 3<br>*Daddy are you?*<br>*Mummy are you?*<br>*Gramps are you?* | 1<br>*Donkey where are you?* | 6<br>*Kaczka are you?* 'duck are you?'<br>*Tata are you?* 'daddy are you?'<br>*Buty are you?* 'shoes are you?'<br>*Kot are you?* 'cat are you?'<br>*Autobus are you?* 'bus are you?'<br>*Daddy kapci are you?* 'daddy slippers are you? | 0 |
| Mine X | 1<br>*Mine book* | 1<br>*Mine spoon* | 1<br>*Mine piłka* 'mine ball' | 0 |
| Moje X 'My X' | 2<br>*Moje ciasto* 'my cake'<br>*Moje jabłko* 'my apple' | 0 | 1<br>*Moje coat* | 0 |
| X('s) turn | 1<br>*Mummy's turn* | 7<br>*Me turn*<br>*My turn*<br>*You turn*<br>*Your turn*<br>*Mummy turn*<br>*Mummy's turn*<br>*Daddy's turn* | 1<br>*Tata's turn* 'daddy's turn' | 1<br>*Tata's turn* 'daddy's turn' |
| X; come | 1<br>*Mummy come!* | 0 | 1<br>*Tata come!* 'daddy come!' | 0 |
| Daj mi X 'Give me X' | 1<br>*Daj mi smoczek* 'dummy' | 1<br>*Daj tata* 'give daddy' (to mother) | 3<br>*Daj mi pots*<br>*Daj mi that*<br>*Daj mi keys* | 6<br>*Daj more*<br>*Daj breakfast*<br>*Daj big one*<br>*Daj mi yours*<br>*Daj mi that*<br>*Daj playdough* |
| X; look | 1<br>*Mummy look!* | 1<br>*Daddy look!* | 1<br>*Tata look!* 'daddy look!' | 1<br>*Tata look!* 'daddy look!' |
| X coming | 2<br>*Mummy coming?*<br>*Car coming?* | 1<br>*Mummy coming?* | 2<br>*Tata coming?* 'Daddy coming?'<br>*Mleko coming?* 'Milk coming?' | 1<br>*Tata coming?* 'Daddy coming?' |
| X śpi 'X is asleep' | 1<br>*Dzidzia śpi* 'Baby's asleep' | 0 | 1<br>*Mummy śpi* 'Mummy's asleep' | 0 |

**Table A2.** *Cont.*

| | | | | |
|---|---|---|---|---|
| (I) want (it) X | 3<br>*I want one*<br>*Want yoghurt*<br>*I want cheese* | 11<br>*I want it fruit*<br>*I want it water*<br>*I want it juice*<br>*I want donkeys*<br>*I want peekaboo*<br>*I want fork*<br>*Want two*<br>*I want two*<br>*I want to eat*<br>*I want more*<br>*I want cut cut cut*<br>*I want this*<br>*I want this one*<br>*I want that one*<br>*I want to play the drum the ding dong* | 1<br>*I want do domu* 'I want home' | 3<br>*I want woda* 'water'<br>*I want it karty* 'cards'<br>*I want it karty* 'cards' *any more* |
| I don't want it X | 1<br>*I don't want it dancing* | 12<br>*I don't want it*<br>*I don't want it one piece*<br>*I don't want it dinner*<br>*I don't want it the garden*<br>*I don't want it fish*<br>*I don't want it playdough*<br>*I don't want it Barbie*<br>*I don't want it garden*<br>*I don't want that*<br>*I don't want cat*<br>*I don't want butterfly*<br>*I don't want stickers* | 4<br>*I don't want it pies* 'dog'; *I don't want it ser* 'cheese'; *I don't want it mleko* 'mleko'; *I don't want it spać* 'to sleep' | 4<br>*I don't want it do domu* 'to home'<br>*I don't want it dom* 'home'<br>*I don't want truskawki* 'strawberries'<br>*I don't want ciasto* 'cake' |
| I need X | 4<br>*I need milk*<br>*I need pasta*<br>*I need juice*<br>*I need head* | 3<br>*Need drums*<br>*I need my torch*<br>*I need the shakers* | 1<br>*I need mleko* 'milk' | 0 |
| I'm gonna X | 1<br>*I'm gonna play Eric* | 2<br>*I'm gonna drawing too*<br>*I'm gonna clean up* | 1<br>*I'm gonna łap* 'catch' *to mummy* | 0 |
| Look; X | 3<br>*Look that*<br>*Look what a mess*<br>*Look bathtime* | 2<br>*Look train*<br>*Look horn* | 2<br>*Look nos*<br>'Look nose!' *Look dzidzia* 'Look baby!' | 0 |
| Everybody X | 7<br>*Everybody look*<br>*Everybody sit down*<br>*Everybody shower*<br>*Everybody up*<br>*Everybody bicycle*<br>*Everybody apple*<br>*Everybody tired* | 4<br>*Everybody stairs*<br>*Everybody like this*<br>*Everybody wake*<br>*Everybody's here* | 1<br>*Everybody ząbki* 'Everybody teeth' | 0 |
| Daddy; X! | 2<br>*Daddy have a go!*<br>*Daddy help Sadie!* | 12<br>*Daddy hat on*<br>*Daddy animals*<br>*Daddy drawing*<br>*Daddy snake*<br>*Daddy wake up everybody*<br>*Daddy look*<br>*Daddy you play this*<br>*Daddy no*<br>*Daddy mummy*<br>*Daddy play*<br>*Daddy tiger*<br>*Daddy back* | 1<br>*Daddy śpij*! 'Daddy sleep!' | 0 |

**Table A2.** *Cont.*

| Tata; X! 'Daddy; X!' | *Tata, do domu* 'daddy, home!' | 0 | *Tata stop!* *Tata come!* *Tata come back! Tata look* | *Tata no!* *Tata look!* |
|---|---|---|---|---|
| Mummy; X | 3 *Mummy go swim* *Mummy hiding?* *Mummy why going?* | 0 | 1 *Mummy daj!* 'mummy give!' | 2 *Mummy daj* 'give' *Mummy daj mi* 'give me' |

## Appendix E

Slot fillers in the most frequently used mostly monolingual constructions (as recorded on video)

(1)  I X

> *Got; found; drop; want; don't X (don't want; don't know); say; gonna; find; not finished; have; need; play; finished; swim; like*

(2)  The X

> *juice; park; ball; garden; flower; trousers; house; beep beep; baby; toilet; lights; train; sheep; drum; shakers; garden; top; baby; cards; mummy; table; wall; jungle; other part; bus; cake; door; one; belly; moon; telephone; tickets; ding dong; twinkle twinkle*

(3)  My X

> *turn; teddy bear; mummy; daddy; book; hat; juice; trousers; torch; tata* 'daddy'; *peepo; baby; breakfast; flowers; dom* 'house'; *Sadie; teeth; raisins; water; koń* 'horse'; *drawing; gosh; everybody's; tickets*

(4)  Where('s) X?

> *other one; are you; daddy; my X (my mummy; my torch; my elephant; my everybody; my cut cut); babcis* 'grandma's'; *the X (the other; the one); more X (more flowers; more rice)*

## Appendix F

Slot fillers in the most frequently used mostly bilingual constructions (as recorded on video)

(1)  More X

> *juice; painting; caterpillar; cereal; milk; fruit; Sadie; mleko* 'milk'; *car; banana; pasta; what; cheese; jeszczes* 'more+s'; *jedens* 'one+s'; *krede*'chalk'; *jeszcze krede* 'more chalk'; *małpa* 'monkey'; *owca* 'sheep'; *pies* 'dog'; *kura*'hen'; *monkeys; zebra; lion; koko; sheep; flowers; book; modelina* 'playdough'; *yoghurt; cake; rice; bread*

(2)  No X

> *airplane; tata* 'daddy'; *daddy; mummy; green apple; yoghurt; potato; potatoes; ziemniaczki* 'potatoes'; *sausage; juice; drinking the juice; more juice; more; one piece; play the park; my turn; hand please; big cat; eating; banging; drawing; drawing picture; working; lion; bath; fingers; hat; like X (like it; like sok* 'juice'); *want (it); wipe it; wake up; eat it; Sadie; kawałek* 'piece'; *fish; rączki* 'hands'; *świnka* 'piggy'; *krowa* 'cow'; *pies* 'dog'; *tutaj* 'here'; *chcę* 'I want'; *ready; put up; miś* 'teddy'; *me; mine; you; take our pencil; touch; help; yellow; Peppa Pig; cheese; jedz* 'you eat!'; *chleba* 'bread+infl'; *sausage; kąpać* 'to bathe'; *seven; truskawkis* 'strawberries'; *Lailani birthday; Sadie birthday; dziękuję* 'thank you'; *orange; pen; hungry; snail; careful*

(3)  X gone

> *butterfly; all; pasta; mama; ice-cream; a baby; dzidzia* 'baby'

(4)  Daj (mi) X 'Give me X'

> *more; breakfast; big one; yours; that*

(5)  (I don't) want (it) X

> *tata* 'daddy'; *one piece; fruit; water; juice; dinner; donkeys; the garden; garden; do domu* 'in the direction of home'; *dom* 'house'; *fish; playdough; Barbie; peekaboo; Sadie; fork; woda* 'water'; *two; two kaczki* 'ducks'; *to eat my breakfast; more; truskawki* 'strawberries'; *cat; butterfly; stickers; ciasto* 'cake'; *cut cut cut; this; this one; that; that one; to play the drum; karty* 'cards' *(any more)*

(6)  (Where) other (one) X?

> *tattoo; pies* 'dog'; *kot* 'cat'; *cat; kura* 'hen'; *królik* 'rabbit'; *zebra; zwierzęta* 'animals'; *zwierząt* 'animals+infl; *Incy Wincy spider; monkey; fish; sheep; owca* 'sheep'; *owcas* 'sheep+plural'; *misiu* 'teddy'; *kangaroos; penguin; butterfly; butterflies; hopping; part; down; auto* 'car'; *ding dong; cake; stickers; tiger; kaczki* 'ducks'; *kaczkis* 'ducks+English plural'; *konia* 'horse'; *fork; spoon*

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
