# Peer review of "Slot-and-Frame Schemas in the Language of a Polish- and English-Speaking Child: The Impact of Usage Patterns on the Switch Placement"

_languages, doi:10.3390/languages4010008_

Round 1

Reviewer 1 Report

The aim of the paper is to evaluate the characteristics and typology of slot and frame schemas in the simultaneous child bilingual language production. It focuses on the impact of language use and dominance on the patterns of code-switching in the production of the schemas.

The results will contribute to the field of simultaneous bilingual language acquisition in young children, specifically the areas of code-switching, language use and dominance.

 I recommend the paper for publication pending the following revisions:

I would recommend that the reference list is updated to include more current publications, especially in the area of language dominance and code-switching. The literature for, e.g. translation equivalents refers to years 1995-2000, CS in parental input 1988-2003, language dominance 1998. Please see current work by Nicoladis, Costa, Blom, Gathercole, Hoff, among others, in order to update the above.

There are some referents missing in the text, please update, e.g. lines 83-84, 392-394, to avoid claims without proof in the literature.

The issue of language dominance: I do not think that describing the language the child used more often as her preferred language is evidence based. How was the apparent preference evaluated? In the current literature, language use is typically linked to language dominance (e.g. Lanza, de Bot, Kopke, etc) which is a psycholinguistic phenomenon, and language preference is viewed to be more of a result of the interface between language use and dominance. Therefore, I would recommend using the stronger/dominant language term. However, this brings on another question of how was language preference /dominance measured in the context of this study.

In the light of the above, I would recommend to make more links between the issue of language input and dominance from the psycholinguistic perspective.

Data collection: please explain who was responsible for updating the diary when the mother worked

The structure of the paper: logical and coherent. I would recommend structuring the introduction into more sections to include aims of the study, research questions for a clearer communication of ideas.

Communication of ideas: clear and coherent

The language: clear, however, the choice of some nouns seems questionable and not within the expected range, e.g. the protagonist of the study, please change for subject or participant; the birth of syntax, etc.

Overall interesting results presented in a concise and relevant manner. 

Author Response

Thank you for your comments on our paper and please see our responses in bold.

I would recommend that the reference list is updated to include more current publications, especially in the area of language dominance and code-switching. The literature for, e.g. translation equivalents refers to years 1995-2000, CS in parental input 1988-2003, language dominance 1998.

Please see current work by Nicoladis, Costa, Blom, Gathercole, Hoff, among others, in order to update the above. Thank you for pointing this out. We find that the work of Costa and Blom is mostly concerned with the issue of code-switching vs. executive functions. However, we have introduced the work of Paradis & Nicoladis (2007); Cantone (2007); Legacy, Pascal, Friend and Poulin-Dubois (2016); Comeau, Genesee and Lapaquette (2003); Mishina-Mori (2011); Unsworth (2015); Nicoladis, Hui and Wiebe (2018); Mueller et al (2015). See lines 176-222

There are some referents missing in the text, please update, e.g. lines 83-84, 392-394, to avoid claims without proof in the literature. Lines 83-84 produce an overarching statement which is supported in lines 85 and 89 where we refer to two specific studies which provide evidence for it. We have added a reference in lines 392-94.

The issue of language dominance: I do not think that describing the language the child used more often as her preferred language is evidence based. How was the apparent preference evaluated? In the current literature, language use is typically linked to language dominance (e.g. Lanza, de Bot, Kopke, etc) which is a psycholinguistic phenomenon, and language preference is viewed to be more of a result of the interface between language use and dominance. Therefore, I would recommend using the stronger/dominant language term. However, this brings on another question of how was language preference /dominance measured in the context of this study. A fair point, thank you. We have changed ‘preferred’ to ‘dominant’. Sadie’s dominance in English was established using several measures, as explained in the original version of the article. Please see lines 255-265.

In the light of the above, I would recommend to make more links between the issue of language input and dominance from the psycholinguistic perspective. Both input and output are linked to dominance in lines 255-265.

Data collection: please explain who was responsible for updating the diary when the mother worked Line 259: No diary entries were made when Sadie’s mother was at work.

The structure of the paper: logical and coherent. I would recommend structuring the introduction into more sections to include aims of the study, research questions for a clearer communication of ideas. See line 117: The introduction has been broken up into three. The second section has been named ‘Our research questions’. The third ‘Our contribution to bilingual research’

The language: clear, however, the choice of some nouns seems questionable and not within the expected range, e.g. the protagonist of the study, please change for subject or participant; the birth of syntax, etc. Both corrected to ‘participant of the study’ and ‘emergence of syntax’

Reviewer 2 Report

This paper presents the results of an experiment on language acquisition by one bilingual child (Sadie; English and Polish). The theoretical framework is the emergence of 'slot-and-frame' schemas from a usage-based point of view. I think that the study is well grounded and developed. The paper is well organized (general introduction, methodology, results, discussion, and conclusion) and well written. I make only a couple of suggestions and I indicate a few typos I noticed.

I suggest dividing the introduction in, at least, a couple of subsections. The first part of the introduction gives some explanation about the 'slot-and-frame' approach; the second part is devoted to introduce the three research questions of the current work and to emphasise the relevance of studies based on bilingual individuals.

Maybe in section 4 a brief note on the properties of the definite article in Polish would be convenient to reinforce the discussion on the 'The X' construction/frame. Some properties of the definite article in French and in German (compared to English) are given, but what about (the absence of) definite articles in Polish? Might this (absence, definite/indefinite interprtaion) be linked to the absence of CS in 'The X'?

I think that the following expressions should be in italics in the text:

- line 502, 'more' in "... find more on its own and ..."

- line 513, 'the' in "... aware that the is a separate ..."

- line 538, 'The X' in "... processing of The X in English ..."

Author Response

Thank you for your comments on our article. Please see our responses in bold below.

I suggest dividing the introduction in, at least, a couple of subsections. The first part of the introduction gives some explanation about the 'slot-and-frame' approach; the second part is devoted to introduce the three research questions of the current work and to emphasise the relevance of studies based on bilingual individuals. The introduction has been broken up into three. The second section has been named ‘Our research questions’. The third ‘Our contribution to bilingual research’

Maybe in section 4 a brief note on the properties of the definite article in Polish would be convenient to reinforce the discussion on the 'The X' construction/frame. Some properties of the definite article in French and in German (compared to English) are given, but what about (the absence of) definite articles in Polish? Might this (absence, definite/indefinite interpretation) be linked to the absence of CS in 'The X'? We considered this explanation but we feel this does not help us to understand the problem.  If there had been some switching within The X in the German-English corpus we quoted, we could have used the argument of structural equivalence between English-German and lack of equivalence between English-Polish. However, the German-English child did not switch within The X, at least not at the ages 2-3 which suggests that the properties of the German construction are of little consequence for the usage of its English equivalent at that developmental stage.

I think that the following expressions should be in italics in the text: (all changed – thank you for pointing this out)

- line 502, 'more' in "... find more on its own and ..."

- line 513, 'the' in "... aware that the is a separate ..."

- line 538, 'The X' in "... processing of The X in English ..."